# Whole genome sequencing of *Plasmodium vivax* isolates reveals frequent sequence and structural polymorphisms in erythrocyte binding genes

**Anthony Ford**[1,2☯]*, **Daniel Kepple**[2☯], **Beka Raya Abagero**[3], **Jordan Connors**[1], **Richard Pearson**[4], **Sarah Auburn**[5], **Sisay Getachew**[6,7], **Colby Ford**[1], **Karthigayan Gunalan**[8], **Louis H. Miller**[8], **Daniel A. Janies**[1], **Julian C. Rayner**[9], **Guiyun Yan**[10]*, **Delenasaw Yewhalaw**[3], **Eugenia Lo**[2]*

1 Department of Bioinformatics and Genomics, University of North Carolina at Charlotte, United States of America, 2 Department of Biological Sciences, University of North Carolina at Charlotte, United States of America, 3 Tropical Infectious Disease Research Center, Jimma University, Ethiopia, 4 Malaria Programme, Wellcome Trust Sanger Institute, Hinxton, United States of America, 5 Global and Tropical Health Division, Menzies School of Health Research and Charles Darwin University, Darwin, Northern Territory, Australia, 6 College of Natural Sciences, Addis Ababa University, Ethiopia, 7 Armauer Hansen Research Institute, Addis Ababa, Ethiopia, 8 Laboratory of Malaria and Vector Research, NIAID/NIH, Bethesda, United States of America, 9 Department of Clinical Biochemistry, Cambridge Institute for Medical Research, University of Cambridge, Cambridge CB2 OXY, United Kingdom, 10 Program in Public Health, University of California at Irvine, United States of America, ☯ These authors contributed equally to this work

☯ These authors contributed equally to this work.
* aford29@uncc.edu (AF); guiyuny@uci.edu (GY); eugenia.lo@uncc.edu (EL)

**Data Availability Statement:** All data has been deposited in the European Nucleotide Archive and the accession numbers are in Table 1.

## Abstract

*Plasmodium vivax* malaria is much less common in Africa than the rest of the world because the parasite relies primarily on the Duffy antigen/chemokine receptor (*DARC*) to invade human erythrocytes, and the majority of Africans are Duffy negative. Recently, there has been a dramatic increase in the reporting of *P. vivax* cases in Africa, with a high number of them being in Duffy negative individuals, potentially indicating *P. vivax* has evolved an alternative invasion mechanism that can overcome Duffy negativity. Here, we analyzed single nucleotide polymorphism (SNP) and copy number variation (CNV) in Whole Genome Sequence (WGS) data from 44 *P. vivax* samples isolated from symptomatic malaria patients in southwestern Ethiopia, where both Duffy positive and Duffy negative individuals are found. A total of 123,711 SNPs were detected, of which 22.7% were nonsynonymous and 77.3% were synonymous mutations. The largest number of SNPs were detected on chromosomes 9 (24,007 SNPs; 19.4% of total) and 10 (16,852 SNPs, 13.6% of total). There were particularly high levels of polymorphism in erythrocyte binding gene candidates including merozoite surface protein 1 (*MSP*1) and merozoite surface protein 3 (*MSP*3.5, *MSP*3.85 and *MSP*3.9). Two genes, *MAEBL* and *MSP*3.8 related to immunogenicity and erythrocyte binding function were detected with significant signals of positive selection. Variation in gene copy number was also concentrated in genes involved in host-parasite interactions, including the expansion of the Duffy binding protein gene (*PvDBP*) on chromosome 6 and *MSP*3.11 on chromosome 10. Based on the phylogeny constructed from the whole genome

**Funding:** This research was funded by National Institutes of Health (NIH R15 AI138002 to EL; NIH U19 AI129326 to GY; NIH R01 AI050243 to GY; D43 TW001505 to GY) and The Wellcome Trust 206194/Z/17/Z to JR. The funders had no role in study design, data collection and analysis, decision to publish, or preparation of the manuscript.

**Competing interests:** The authors have declared that no competing interests exist.

sequences, the expansion of these genes was an independent process among the *P. vivax* lineages in Ethiopia. We further inferred transmission patterns of *P. vivax* infections among study sites and showed various levels of gene flow at a small geographical scale. The genomic features of *P. vivax* provided baseline data for future comparison with those in Duffy-negative individuals and allowed us to develop a panel of informative Single Nucleotide Polymorphic markers diagnostic at a micro-geographical scale.

## Author summary

*Plasmodium vivax* is the most geographically widespread parasite species that causes malaria in humans. Although it occurs in Africa as a member of a mix of *Plasmodium* species, *P. vivax* is dominant in other parts of the world outside of Africa (e.g., Brazil). It was previously thought that most African populations were immune to *P. vivax* infections due to the absence of Duffy antigen chemokine receptor (*DARC*) gene expression required for erythrocyte invasion. However, several recent reports have indicated the emergence and potential spread of *P. vivax* across human populations in Africa. Compared to Southeast Asia and South America where *P. vivax* is highly endemic, data on polymorphisms in erythrocyte binding gene candidates of *P. vivax* from Africa is limited. Filling this knowlege gap is critical for identifying functional genes in erythrocyte invasion, biomarkers for tracking the *P. vivax* isolates from Africa, as well as potential gene targets for vaccine development. This paper examined the level of genetic polymorphisms in a panel of 43 potential erythrocyte binding protein genes based on whole genome sequences and described transmission patterns of *P. vivax* infections from different study sites in Ethiopia based on the genetic variants. Our analyses showed that chromosomes 9 and 10 of the *P. vivax* genomes isolated in Ethiopia had the most high-quality genetic polymorphisms. Among all erythrocyte binding protein gene candidates, the merozoite surface proteins 1 and merozoite surface protein 3 showed high levels of polymorphism. MAEBL and MSP3.8 related to immunogenicity and erythrocyte binding function were detected with significant signals of positive selection. The expansion of the Duffy binding protein and merozoite surface protein 3 gene copies was an independent process among the *P. vivax* lineages in Ethiopia. Various levels of gene flow were observed even at a smaller geographical scale. Our study provided baseline data for future comparison with *P. vivax* in Duffy negative individuals and help develop a panel of genetic markers that are informative at a micro-geographical scale.

## Introduction

Vivax malaria is the most geographically widespread human malaria, causing over 130 million clinical cases per year worldwide [1]. *Plasmodium vivax* can produce dormant liver-stage hypnozoites within infected hosts, giving rise to relapse infections from months to years. This unique feature of *P. vivax* contributes to an increase in transmission potential and increases the challenge of elimination [2]. Understanding *P. vivax* genome variation will advance our knowledge of parasite biology and host-parasite interactions, as well as identify potential drug resistance mechanisms [3, 4]. Such data will also help identify molecular targets for vaccine development [5–7] and provide new means to track the transmission and spread of drug resistant parasites [8–9].

Compared to *P. falciparum*, *P. vivax* isolates from Southeast Asia (e.g., Thailand and Myanmar), Pacific Oceania (Papua New Guinea), and North and South America (Mexico, Peru, and Colombia) have significantly higher nucleotide diversity at the genome level [2]. This could be the result of frequent gene flow via human movement, intense transmission, and/or variation in host susceptibility [10–14]. *P. vivax* infections are also much more likely to contain multiple parasite strains in areas where transmission is intense and/or relapse is common [10, 15–18]. In Papua New Guinea, for example, *P. vivax* infections had an approximately 3.5-fold higher rate of polyclonality and nearly double the multiplicity of infection (MOI) than the *P. falciparum* infections [16]. Similar rates of polyclonality and MOI have also been reported in *P. vivax* in Cambodia [6]. It is possible intense transmission has sustained a large and stable parasite population in these regions [17,18]. In contrast, geographical differentiation and selection pressure over generations can lead to fixation of parasite genotypes in local populations. In the Asia-Pacific, *P. vivax* showed a high level of genetic relatedness through inbreeding among the dominant clones, in addition to strong selection imposed in a number of antimalarial drug resistance genes [19]. In Ethiopia, the chloroquine resistance transporter gene (*Pvcrt*) of *P. vivax* on chromosome 14 had been shown with significant selection in a region upstream of the promotor, highlighting the ability of *P. vivax* to rapidly evolve in response to control measures [20]. Apart from mutations, high copy number observed in *Pvcrt* and the multidrug resistant gene (*Pvmdr*1) has also been shown to be associated with increased antimalaria drug resistance [21,22].

Recent genomic studies have indicated that some highly polymorphic genes in the *P. vivax* genome are associated with red blood cell invasion and immune evasion [10, 12, 19, 23]. They include the merozoite surface protein genes *MSP*1 (PVP01_0728900) and *MSP*7 (PVP01_1219700), the reticulocyte binding protein gene *RBP*2c (PVP01_0534300), and serine-repeat antigen 3 (*SERA*; PVP01_0417000) [23–29]. Polymorphisms in genes associated with immune evasion and reticulocyte invasion have important implications for the invasion efficiency and severity of *P. vivax* infections. Members of the erythrocyte binding gene family, including reticulocyte binding proteins (*RBP*s), Duffy-binding proteins (*DBP*s), and merozoite surface proteins (*MSP*3 and *MSP*7) have been previously shown to exhibit high sequence variation in *P. vivax* [20, 30]. The polymorphisms in *RBP*1 and *RBP*2 genes may relate to an increased capability of erythrocyte invasion by *P. vivax* [31–33]. It has been suggested that Pv*RBP*2b-TfR1 interaction is vital for the initial recognition and invasion of host reticulocytes [34], prior to the engagement of *PvDBP1* and *DARC* and formation of a tight junction between parasite and erythrocyte [35]. Apart from Pv*RBP*, Reticulocyte Binding Surface Antigen (Pv*RBSA*) [36], an antigenic adhesin, may also play a key role in *P. vivax* parasites binding to target cells, possessing the capability of binding to a population of reticulocytes with a different Duffy phenotype [37, 38]. Another erythrocyte binding protein gene (Pv*EBP*), a paralog of *PvDBP1*, which harbors all the hallmarks of a *Plasmodium* red blood cell invasion protein, including conserved Duffy-binding like and C-terminal cysteine-rich domains [39], has been recently shown to be variable in copy number in the Malagasy *P. vivax* [39]. Functional analyses indicated that region II of this gene bound to both Duffy-positive and Duffy-negative reticulocytes, although at a lower frequency compared to *PvDBP*, suggestive of its role in erythrocyte invasion [40]. Both Pv*EBP*1 and Pv*EBP*2 genes exhibit high genetic diversity and are common antibody binding targets associated with clinical protection [41, 42]. Other proteins such as tryptophan-rich antigen gene (*TRAg*), anchored micronemal antigen (*GAMA*), and Rhoptry neck protein (*RON*) have also been suggested to play a role in red cell invasion, especially in low-density infections [43–47]. Information of the polymorphisms in these genes will have important implications on the dynamics of host-parasite interactions.

Compared to Southeast Asia and South America where *P. vivax* is highly endemic, data on polymorphisms in erythrocyte binding gene candidates of *P. vivax* from Africa is limited. Filling this knowlege gap is critical for identifying functional genes in erythrocyte invasion, biomarkers for tracking the African *P. vivax* isolates, as well as potential gene targets for vaccine development. It was previously thought that most African populations were immune to *P. vivax* infections due to the absence of *DARC* gene expression required for erythrocyte invasion. However, several recent reports have indicated the emergence and potential spread of *P. vivax* across Africa [32, 48–50]. The objective of this study is to describe genomic variation of *P. vivax* from Ethiopia. Specifically, we examined the level of genetic polymorphisms in a panel of 43 potential erythrocyte binding protein genes that have been suggested to play a role in the parasite-host invasion process. In addition, we inferred transmission patterns of *P. vivax* infections from different study sites based on the genetic variants. A recent study by Auburn *et al.* [20] has compared the genetic variants of *P. vivax* from Ethiopia with other geographical isolates. In the present study, we focus on the genomic characteristics of *P. vivax* among different study sites in Ethiopia with the goals to establish a baseline for genome comparison with the Duffy-negative *P. vivax* in our ongoing investigation, as well as to develop a panel of Single Nucleotide Polymorphic (SNP) markers informative at a micro-geographical scale.

## Materials and methods

### Ethics statement

Scientific and ethical clearance was obtained from the Institutional Scientific and Ethical Review Boards of Jimma and Addis Ababa Universities in Ethiopia, and The University of North Carolina, Charlotte, USA. Written informed consent/assent for study participation was obtained from all consenting heads of households, parents/guardians (for minors under 18 years old), and each individual who was willing to participate in the study.

### Study area and sample collection

Genomic DNA was extracted from 22 clinical samples collected in Jimma, southwestern Ethiopia during peak transmission season (September–November 2016; Fig 1). Finger-pricked blood samples were collected from individuals with symptoms suspected of malaria infection (who has fever with axillary body temperature > 37.5˚C and with confirmed asexual stages of malaria parasite based on microscopy) or febrile patients visiting the health centers or hospitals at each of the study sites. Thick and thin blood smears were prepared for microscopic examination, and 4–6 ml of venous blood were collected from each *P. vivax*-confirmed patient in K2 EDTA blood collection tubes. For the whole blood samples, we used the Lymphoprep/Plasmodpur-based protocol to deplete the white blood cells and enrich the red blood cell pellets [51]. DNA was then extracted from approximately 1 ml of the red blood cell pellets using Zymo Bead Genomic DNA kit (Zymo Research) following the manufacturer's procedures. The extracted DNA were first assessed by nested and quantitative PCR methods to confirm and quantify *P. vivax* of the infected samples [52]. For these 22 samples, we performed microsatellite analyses using seven different loci [53]. Only monoclonal samples were selected and proceeded for sequencing. Whole genome sequencing was conducted on the Illumina HiSeq 3000 Sequencing Platform at the Wellcome Sanger Institute (European Nucleotide Archive [ENA] accession number of each sample in Table 1). The generated sequence reads were mapped individually to the publicly available reference genome PvP01 from Gene DB using BWA-MEMv2 [54–56]. Only reads that were mapped to *P. vivax* were included. The percentage coverage of the *P. vivax* reads in our samples were high enough to not affect the results.

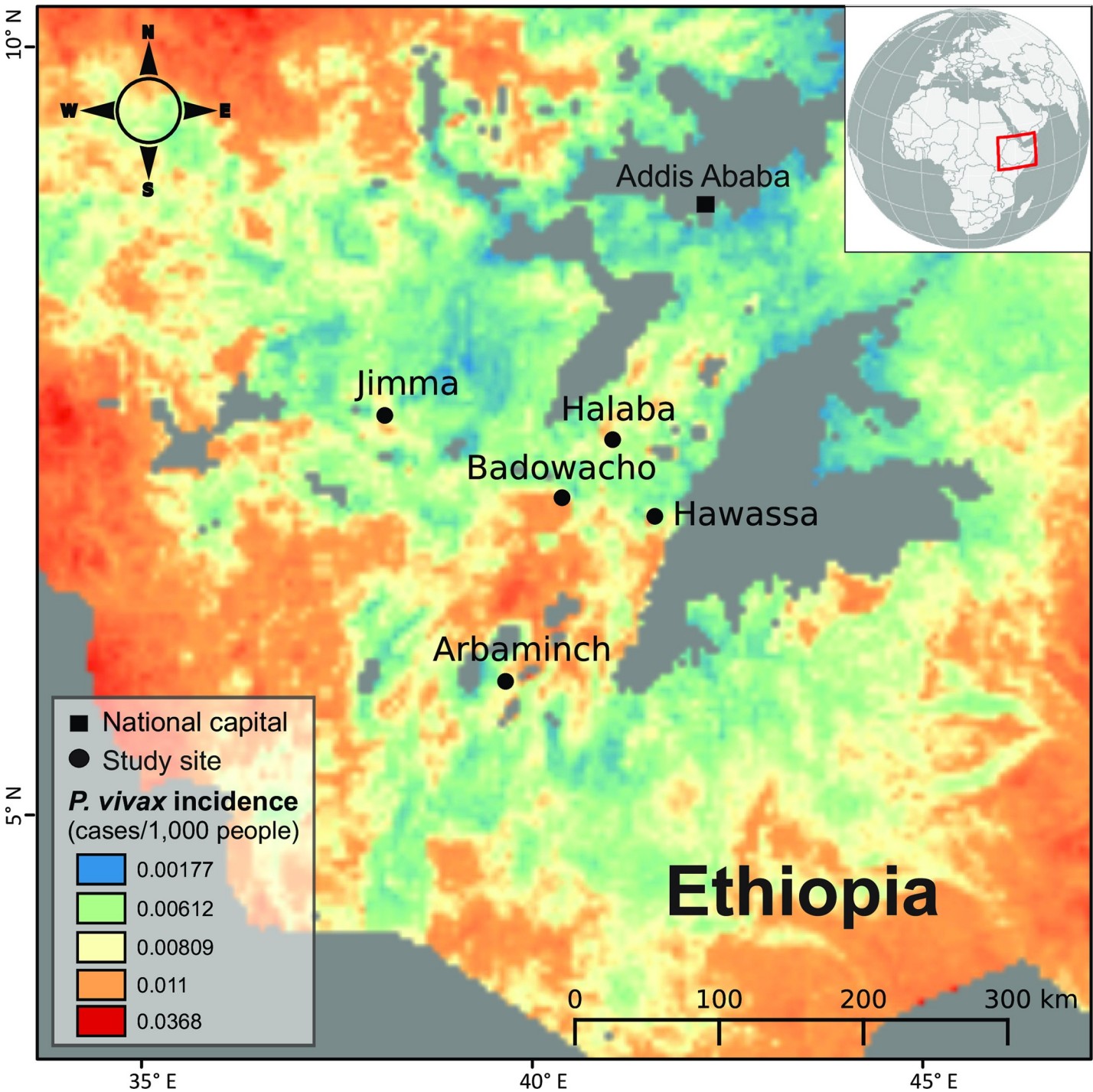

**Fig 1. An overview of the *P vivax* sample collection locations including Arbaminch, Badowacho, Hawassa, Halaba, and Jimma in southwestern Ethiopia.**

An additional 24 sample sequence data were obtained as FASTQ files from the ENA. These samples were collected from Arbaminch, Badowacho, Halaba, and Hawassa in southwestern Ethiopia (Fig 1), the Duffy status of each of these 24 samples is unknown. All *P. vivax* genomes in this study were aligned to the PVP01 reference genome using BWA-MEMv.2 with default

**Table 1. Information of whole genome sequences of 44 *Plasmodium vivax* isolates from Ethiopia.** The European Nucleotide Archive (ENA) accession number for all files. Asterisks indicated genomes generated in this study.

| Sample ID | Study site | ENA accession number | All reads | *P. vivax* reads (%) | Average genome coverage |
|---|---|---|---|---|---|
| BBH(1)-125* | Jimma | ERS849384 | 16104099 | 14765167 (92%) | 83.11 |
| BBH(1)-132* | Jimma | ERS2593850 | 295443478 | 269499723 (91%) | 582.99 |
| BBH(1)-137* | Jimma | ERS2593862 | 15428892 | 12612040 (82%) | 41.09 |
| BBH(1)-153* | Jimma | ERS2593851 | 19596872 | 17206221 (88%) | 57.34 |
| BBH(1)-162* | Jimma | ERS2593852 | 322082372 | 32342978 (10%) | 72.36 |
| HT(1)-144* | Jimma | ERS2593853 | 16508858 | 8025807 (49%) | 28.52 |
| HT(1)-147* | Jimma | ERS2593863 | 118998744 | 97833898 (82%) | 234.14 |
| HT(2)-112* | Jimma | ERS2593864 | 44338048 | 23619451 (53%) | 73.45 |
| JHC(1)-208* | Jimma | ERS2593865 | 381752146 | 294998614 (77%) | 112.34 |
| JHC(2)-100* | Jimma | ERS2593855 | 17622060 | 15958894 (91%) | 56.23 |
| MKH(1)-72* | Jimma | ERS2593857 | 17668719 | 14101177 (80%) | 57.11 |
| MKH(2)-71* | Jimma | ERS2593866 | 217327926 | 208538370 (96%) | 715.07 |
| SGH(1)-355* | Jimma | ERS2593858 | 18782232 | 17776334 (95%) | 75.92 |
| SGH(1)-357* | Jimma | ERS2593859 | 82415713 | 51869956 (63%) | 106.63 |
| SGH(1)-331* | Jimma | ERS849385 | 17931182 | 16408183 (92%) | 90.97 |
| SGH(1)-337* | Jimma | ERS2593867 | 16492008 | 7883420 (48%) | 29.05 |
| SGH(1)-358* | Jimma | ERS2593868 | 13579936 | 7433474 (55%) | 27.75 |
| SGH(1)-359* | Jimma | ERS2593869 | 24113148 | 15808023 (66%) | 46.36 |
| SGH(2)-103* | Jimma | ERS2593861 | 33312612 | 29984852 (90%) | 123.54 |
| SGH(2)-108* | Jimma | ERS2593871 | 6209652 | 4333558 (70%) | 16.45 |
| QS0001-C | Badowacho | ERR775189 | 11815238 | 10870019 (92%) | 27.87 |
| QS0002-C | Badowacho | ERR775190 | 7524038 | 6696393 (89%) | 13.7 |
| QS0003-C | Badowacho | ERR775191 | 20904948 | 19441601 (93%) | 61.29 |
| QS0004-C | Badowacho | ERR775192 | 23248006 | 22085605 (95%) | 68.84 |
| QS0011-C | Hawassa | ERR925433 | 10504728 | 9979491 (95%) | 31.58 |
| QS0012-C | Hawassa | ERR925434 | 31067044 | 30135032 (97%) | 99.69 |
| QS0013-C | Hawassa | ERR925435 | 6971112 | 6413423 (92%) | 19.17 |
| QS0014-C | Hawassa | ERR925409 | 9894300 | 9102756 (92%) | 26.21 |
| QS0015-C | Hawassa | ERR925410 | 9991804 | 9292377 (93%) | 24.88 |
| QS0016-C | Hawassa | ERR925411 | 10435496 | 9705011 (93%) | 28.09 |
| QS0018-C | Hawassa | ERR925412 | 9707986 | 9028426 (93%) | 23.48 |
| QS0025-C | Arbaminch | ERR925416 | 8098728 | 7450829 (92%) | 17.65 |
| QS0027-C | Arbaminch | ERR925417 | 8497142 | 7817370 (92%) | 22.06 |
| QS0028-C | Arbaminch | ERR925436 | 6943513 | 6318597 (91%) | 18.88 |
| QS0031-C | Arbaminch | ERR925437 | 7716892 | 7331047 (95%) | 22.94 |
| QS0032-C | Arbaminch | ERR925420 | 8469946 | 7961749 (94%) | 19.96 |
| QS0033-C | Arbaminch | ERR925421 | 5306456 | 4563552 (86%) | 10.38 |
| QS0035-C | Arbaminch | ERR925438 | 11315474 | 10523390 (93%) | 36.08 |
| QS0037-C | Arbaminch | ERR925439 | 7636696 | 7025760 (92%) | 23.19 |
| QS0042-C | Halaba | ERR925424 | 20411542 | 17145695 (84%) | 10.17 |
| QS0044-C | Halaba | ERR925440 | 20256518 | 18635996 (93%) | 26.98 |
| QS0049-C | Halaba | ERR925441 | 26200128 | 24366119 (93%) | 88.7 |
| QS0051-C | Badowacho | ERR925430 | 6008610 | 4866974 (81%) | 9.22 |
| QS0053-C | Badowacho | ERR925431 | 8790382 | 7999247 (91%) | 23.91 |

settings [55, 56]. The overall quality of each resulting BAM was assessed using FASTQC. Similarly, we concluded that the percentage of the *P. vivax* reads covered in the additional 24 samples were high enough to reflect the dominant signal of the variants and negate polyclonal influences. Two of our samples displayed a significant decline in average quality in read mapping and were therefore removed from further SNP variant and copy number variation analyses.

To provide a comparison of SNP and copy number variants with the Ethiopian isolates, we chose 50 additional *P. vivax* genomes from Southeast Asia (mainly from Cambodia and Thailand where several genomes are published and available) deposited in the ENA database [19]. These sequences were realigned using BWA-MEMv.2, with the same default settings as the Ethiopian genomes, to ensure all map files were constructed using the same tool and parameters. Furthermore, to determine the clonality of the isolates, we calculated the Fws (within-host heterozygosity) statistic using the moimix R package [57]. After performing the variant call, we removed the indels and filtered the SNPs. These high-quality SNPs were then used to calculate the Fws value using the getfws() function following the procedures suggested in the moimix R package.

## SNP discovery, annotation, and filtering

Potential SNPs were identified by SAM tools v.1.6 mpileup procedure [58] in conjunction with BCF tools v.1.6 [58] across all 44 sample BAM files using the PVP01 reference genome. Compared to the Salvador-I, the PVP01 reference genome consists of 14 chromosomal sequences and provides a greater level of gene annotation power and improved assembly of the subtelomeres [56]. We analyzed only sequence reads that were mapped to these 14 major chromosomal sequences. The hypervariable and subtelomeric regions in our samples were discarded during the variant calling procedure and each sample BAM file had duplicates marked using SAMtools 1.6 markdup procedure. For the mpileup procedure, the maximum depth threshold, which determines the number of maximum reads per file at a position, was set to 3 billion to ensure that the maximum amount of reads for each position was not reached. Samples were pooled together using a multisampling variant calling approach. The SNPs were then annotated with SnpEff v.4.3T [59] based on the annotated gene information in GeneDB. Filtering was done using the following standard metrics, including Read Position Bias, Mapping Quality vs Strand Bias, Raw read depth, Mapping Quality Bias, Base Quality Bias, and Variant Distant Bias produced by SAM tools and BCF tools during the variant calling procedure. In Snp Sift, data was filtered by choosing SNPs that had a Phred Quality score $\geq$ 40, a combined depth across samples (DP) $\geq$ 30 based on post variant call, and a base quality bias >0.1 [60]. We then calculated the allele frequency for each SNP position for all 44 samples using the frequency procedure in VCF tools v.0.1.15 [61]. The total number of SNPs across all samples, as well as the number of nonsynonymous and synonymous mutations were recorded. Mutations were compared among the 14 chromosomes in addition to a panel of 43 erythrocyte binding genes. The same analysis was performed for the 50 Southeast Asian *P. vivax* genomes [19].

## Copy number variation analyses

Copy number variation of gene regions was assessed with CNVnator [62]. CNVnator uses mean-shift theory, a partitioning procedure based on an image processing technique and additional refinements including multiple bandwidth partitioning and GC correction [62]. We first calculated the read depth for each bin and correct GC-bias. This was followed by mean-shift based segment partition and signal merging, which employed an image processing technique. We then performed CNV calling, of which segments with a mean RD signal deviating

by at least a quarter from genomic average read depth signal were selected and regions with a *P*-value less than 0.05 were called. A one-sided test was then performed to call additional copy number variants. SAM tools v.1.6 was utilized in our data preprocessing step to mark potential duplicates in the BAM files and followed the CNV detection pipeline [63]. We extracted the read mappings from each of BAM files for all chromosomes. Once the root file was constructed using the extracted reads, we generated histograms of the read depths using a bin size of 100. The statistical significance for the windows that showed unusual read depth was calculated and the chromosomes were partitioned into long regions that have similar read depth.

To validate the results from CNVnator, we used the GATK4 copy number detection pipeline to further examine gene copy number [64–66]. The read coverage counts were first obtained from pre-processed genomic intervals of a 1000-bp window length based on the PvP01 reference genome. The read fragment counts were then standardized using the Denoise Read Counts that involved two transformations. The first transformation was based on median counts, including the $\log_2$ transformation, and the counts were normalized to center around one. In the second transformation, the denoises tool was used to standardized copy ratios using principal component analysis. Copy numbers were calculated for the additional 50 Southeast Asian genomes to compare with the 44 Ethiopian genomes using the same procedures described above.

## Test for positive selection

Regions of positive selection were examined among the 44 *P. vivax* isolates from Ethiopia using the integrated haplotype score approach, specifically the SciKit-Allel for python, a package used for analysis of large-scale genetic variation data [67]. Before the samples were run through Scikit-Allel, genotypes for each of the samples were phased using BEAGLE [68]. Genes that were detected with signals of positive selection by SciKit-Allel, as well as a panel of 43 potential erythrocyte binding genes were further evaluated using the PAML package (Phylogenetic Analysis by Maximum Likelihood) [69]. Using the codeml procedure in PAML, DNA sequences were analyzed with the maximum likelihood approach in a phylogenetic framework. The synonymous and nonsynonymous mutation rates between protein-coding DNA sequences were then estimated in order to identify potential regions of positive selection. We created two models, the neutral model M1 and the selection model M2. The average $d_N/d_S$ values were estimated across all branches in both M1 and M2 models and the average $d_N/d_S$ values across all sites in the M2 model. The $d_N/d_S$ values were compared between the two models using a likelihood ratio test for significant positive selection.

## Comparison of nucleotide diversity among EBP gene regions

Based on the literature [23–33], we identified 43 gene regions that are potentially related to erythrocyte binding in *P. vivax* (S1 Table). These included the *DBP* (duffy binding protein), *EBP* (erythrocyte binding protein), *MSP* (merozoite surface protein), and *RBP* (reticulocyte binding protein) multigene families, the tryptophan rich antigen gene family (*TRAg*), GPI-anchored microanemal antigen (*GAMA)*, microneme associated antigen (*MA*), rhoptry associated adhesin (*RA*), high molecular weight rhoptry protein 3 (*RHOP*3), and rhoptry neck protein (*RON)* genes. Previous study has shown that the transcriptome profiles of the *TRAg* genes were differentially transcribed at the erythrocytic stages, indicating that these genes may play specific roles in blood-stage development [43]. The reticulocyte binding protein multigene family encodes genes that each have a receptor on the surface that is essential for the host-invasion stage of *P. vivax* [70]. The *MSP* multigene family, currently assumed to be a candidate for vaccine generation, also plays a role in the invasion stage of *P. vivax* and is also immunogenic

[26]. The nucleotide diversity of 43 potential erythrocyte binding genes were compared among the 44 Ethiopian and 50 Southeast Asian *P. vivax* sample consensus sequences using DnaSP [71]. The Pairwise-Deletion method where gaps were ignored in each pairwise comparison was used for this calculation.

### Genetic relatedness and transmission network analyses

Phylogenetic analyses were performed to infer the genetic relatedness among the 44 Ethiopian isolates. Sequence alignment was first conducted using a multiple sequence alignment program in MAFFT v. 7 [72]. The alignment was then trimmed to remove gaps using trimal (the *gappyout* option) that trimmed the alignments based on the gap percentage count over the whole alignment. After sequence editing, we concatenated all alignment files using FASconCAT-G [73], a perl program that allows for concatenation and translation (nucleotide to amino acid states) of multiple alignment files for phylogenetic analysis. We used the maximum likelihood method implemented in the Randomized Accelerated Maximum Likelihood (RAxML) v8 to construct phylogenetic trees [74]. The GTRGAMMA model was used for the best-scoring maximum likelihood tree. The GTR model incorporates the optimization of substitution rates and the GAMMA model accounts for rate heterogeneity. A total of 100 rapid bootstrap runs were conducted to evaluate the confidence of genetic relationships. In addition, we performed principal component analyses using the glPCA function in R, a subset of the adegenet package [75], to determine the genetic relatedness of the samples among the different study sites in Ethiopia. A transmission network was created using StrainHub, a tool for generating transmission networks using phylogenetic information along with isolate metadata [76]. The transmission network was generated using the locations of the samples as the nodes and calculating the source hub ratio for each location. The source hub ratio was calculated by the number of transitions originating from a node over the total number of transitions related to that node. A node with a ratio close to 1 indicates a source, a ratio close to 0.5 indicates a hub, and a ratio close to 0 indicates a sink for the *P. vivax* infections. To validate the analyses of Strainhub, we also performed $F_{ST}$ and ADMIXTURE analyses to determine the level of genetic differentiation among samples between the different districts in Ethiopia. The $F_{ST}$ analysis was done using our high-quality SNPs and the VCFtools function weir-fst-pop. We generated the corresponding bed, fam, and.bim files using PLINK v2 [77]. These files were then used in the ADMIXTURE analysis [78]. ADMIXTURE uses the same statistical model as the STRUCTURE software, and calculates the estimates using a numerical optimization algorithm [78]. We performed the ADMIXTURE analysis from 1 (no genetic differentiation among all study sites) to five (all study sites were genetically differentiated) genetic clusters (*K*) and recorded the respective cross-validation scores. The optimal *K* was selected based on the smallest error rate and lowest cross-validation scores [78].

## Results

### Distribution of SNPs among the chromosomes and EBP genes

A total of 123,711 SNPs was detected among the 44 *P. vivax* samples from Ethiopia (Fig 2), with 22.7% (28,118 out of 123,711) nonsynonymous and 77.3% (95,593 out of 123,711) synonymous mutations (Fig 3A). The highest number of high quality SNPs were observed on chromosomes 9 (24,007 SNPs; 19.4% of the total SNPs), 10 (16,852 SNPs; 13.6%), and 4 (11,354 SNPs; 9.2%); whereas the lowest number of SNPs were observed on chromosomes 2 (1,912 SNPs; 1.5%), 6 (2,977 SNPs; ~0.2%), and 7 (4,779 SNPs; 3.9%; Fig 3A; S2 Table). The number of high-quality SNPs on each chromosome was not shown to be dependent on the size of the chromosome. For the isolates from Southeast Asia, 413,873 variants were detected after

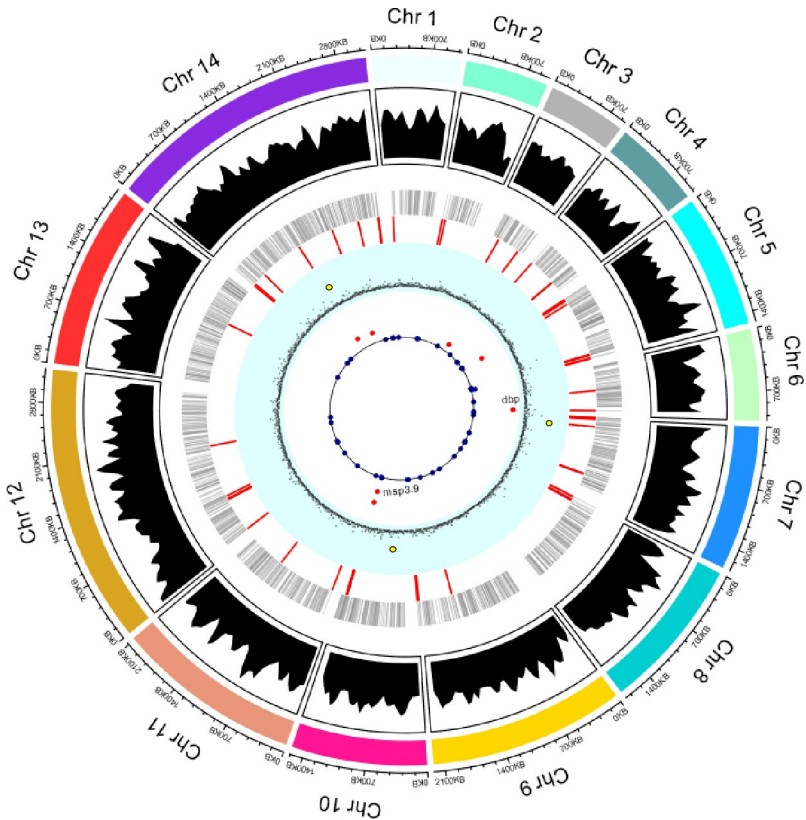

**Fig 2. A summary representation of the *P. vivax* genome, with the outer ring as an ideogram representing the 14 nuclear chromosomes and sizes of each.** The second track represented the average coverage for each chromosome among the 44 Ethiopian samples. The third track containing the gray vertical dashes represented the distribution of genes across the 14 chromosomes. The fourth track that contained the red vertical lines represented the 43 erythrocyte binding gene candidates. The fifth inner track with the light blue background represented the $d_N/d_S$ ratio calculated by partitioning the chromosomes into genomic regions and $d_N/d_S$ directly. The three outliers (yellow dots) represented three unknown plasmodium protein genes that were detected with significant positive selection. The sixth track indicated the overall copy number variation calculated using CNVnator. Red dots represented genes with copy number variation among the Ethiopian genomes.

performing filtration, with 73,547 nonsynomymous (17.77%) and 340,326 synonymous (82.22%). Similar to the isolates from Ethiopia, chromosome 9 of the isolates from Southeast Asia had the largest number of SNPs (66,834 SNPs; 16.1% of the total variants), with 11% being nonsynonymous and 89% synonymous. This was followed by chromosomes 10 of the isolates from Southeast Asia (48,425 SNPs; 11.7%) with 24% nonsynonymous and 76% synonymous substitutions, and chromosome 4 (41,362 SNPs; 9.9%) with 22% nonsynonymous and 78% synonymous substitutions (Fig 3B). The number of detected SNPs on these chromosomes was much higher in the Southeast Asian than the Ethiopian isolates, given similar sample size.

For the isolates fom Ethiopia, the 43 erythrocyte binding genes accounted for 2,361 of the total SNPs, with 1,087 (46%) identified as nonsynonymous and 1,274 (54%) as synonymous mutations (Fig 3C; S1 Table). Among these genes, the highest number of SNPs were observed in the *MSP*3 multigene family (*MSP*3.5, *MSP*3.9 and *MSP*3.8) on chromosome 10 from the isolates from Ethiopia, with an average nucleotide diversity of 2.8%, respectively among our samples (Fig 3B). By contrast, the lowest number of SNPs detected among all the 43 erythrocyte binding genes were the Duffy binding protein gene (*DBP*1) on chromosome 6 from the isolates from Ethiopia with a total of 13 SNPs, of which 12 were identified as nonsynonymous and one

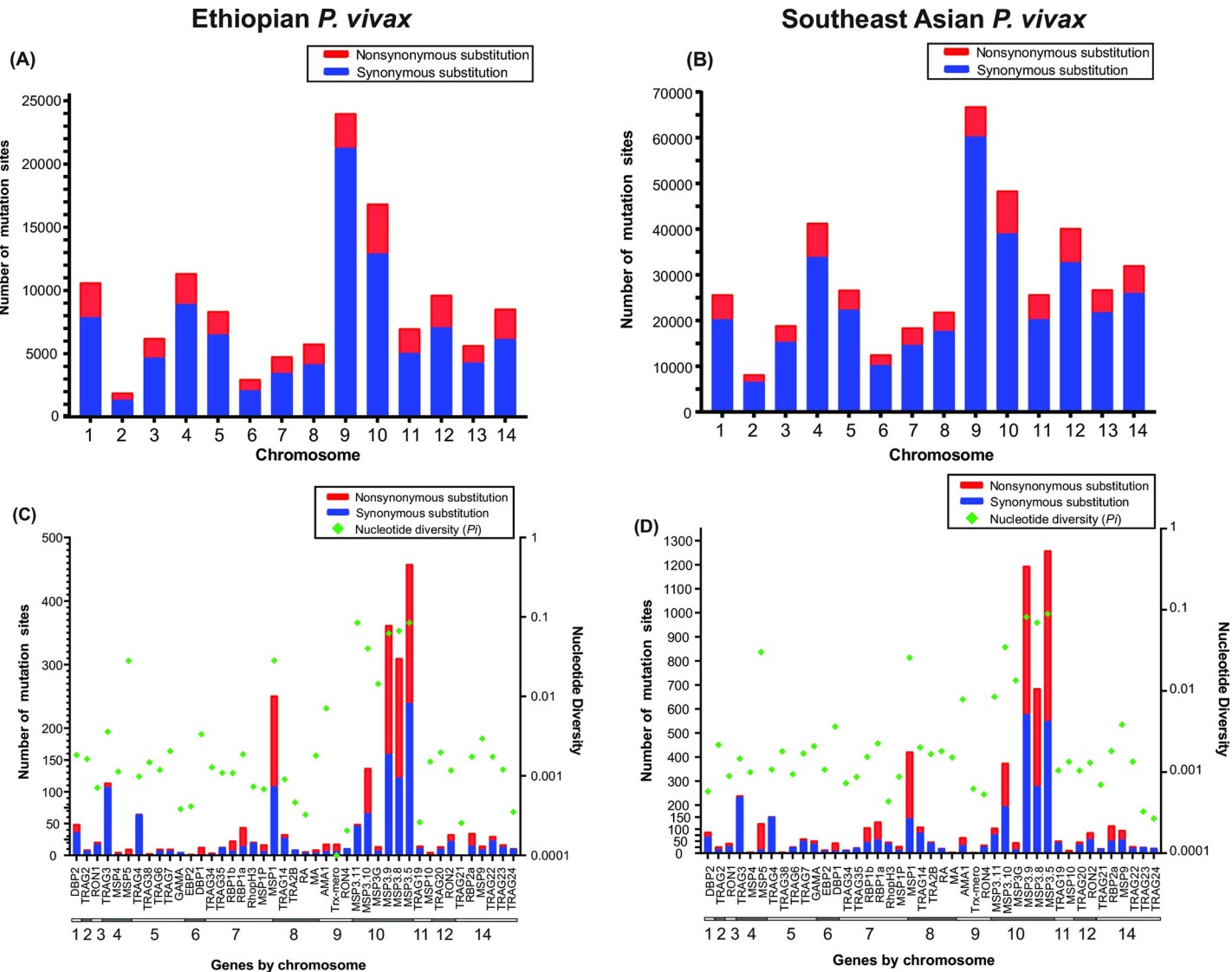

**Fig 3.** (A & B) Distribution of the nonsynonymous and synonymous mutations of each chromosome among the 44 *P. vivax* genomes from Ethiopia and 50 *P. vivax* genomes from Southeast Asian. A higher proportion of synonymous mutations was observed compared to nonsynonymous mutations. Chromosomes 7, 9, and 12 have the most mutations overall, with chromosomes 6 and 3 having the fewest number of mutations in the *P. vivax* genomes from Ethiopia. (C & D) Number of mutation sites and the nucleotide diversity of 43 erythrocyte binding genes among the 44 isolates of *P. vivax* from Ethiopia and 50 genomes of *P. vivax* from Southeast Asia. The *PvMSP* multigene family has the highest number of polymorphic sites when compared to the others, with *PvMSP*3 and *PvMSP*1 the highest number of nonsynonymous and synonymous mutations. Approximately 40% of the mutations were nonsynonymous. These genes were also indicated with the highest nucleotide diversity.

as synonymous mutations (Fig 3B). Likewise, another erythrocyte binding protein (*EBP*2), located also on chromosome 6 from the isolates from Ethiopia, was one of the least variable genes with only one nonsynonymous mutation. The *TRAg* gene family from the isolates from Ethiopia also showed a low level of nucleotide diversity when compared to the other *EBP* gene families with an average nucleotide diversity of 0.2% (Fig 3B).

For the Southeast Asian isolates, the 43 erythrocyte binding genes accounted for 6,130 total SNPs, with 2,923 (47.7%) nonsynonymous and 3,207 (52.3%) synonymous, a ratio similar to the Ethiopian isolates (Fig 3D). The nucleotide diversity was highest in the *MSP* gene family with a nucleotide diversity of 8.8% in *MSP*3.5 and *MSP*3.9 with 8.2%, followed by 6.9% in

*MSP*3.8. On the other hand, the lowest nucleotide diversity was observed in the *TRAg* gene family (0.03% in *TRAg*23 and 0.03% in *TRAg*24), followed by *RHOPH*3 (0.04%) and *RON*4 (0.05%; Fig 3D).

## Estimates of polyclonality based on Fws statistics

The moimix analyses showed that 24 of the 44 isolates from Ethiopia had Fws values of more than or equal to 0.90 with the highest values in 10 samples from Jimma, five samples from Badowacho, five samples from Arbaminch, two samples from Halaba, and the remaining two samples from Hawassa (S3 Table). Eight of the isolates from Ethiopia were shown to have low Fws values ranging from 0.4 to 0.7. They included isolates from Arbaminch (Fws value = 0.423) followed by a isolates from Jimma (Fws value = 0.512), indicative of the presence of more than one clone within the sample (S3 Table). For these eight isolates from Ethiopia, the SNPs and copy number variants were analyzed with the major clone only.

## Gene regions under positive selection

Based on the integrated haplotype scores of our Ethiopian isolates, positive selection was detected in two gene regions (Fig 4). These included the membrane associated erythrocyte binding-like protein (*MAEBL*) on chromosome 9 as well as *MSP*3.8 on chromosome 10 (Fig 4). Based on PAML, 17 out of the 43 erythrocyte binding genes showed evidence of positive selection (Table 2; S4 Table). The majority of these genes belong to the *TRAg* multigene family. The *TRAg* genes had an average $d_N/d_S$ ratio of 2.75 across all branches and an average of 5.75 across all sites for the M2 model tested for selection (Table 2). Compared to the other *TRAg* genes, *TRAg*15 had more sites detected under positive selection, with 50 of the sites showing a posterior probability greater than 50% and 43 showing a posterior probability greater than 95% (Table 2). While the *TRAg*4 gene had the highest $d_N/d_S$ ratio across all sites among other *TRAg* genes, only six sites were shown under positive selection with a posterior probability greater than 50% and one with a posterior probability greater than 95%.

   Apart from the *TRAg*4 genes, *RBP*1a and *RBP*2a belong to the *RBP* gene family also showed significant signals of positive selection (average $d_N/d_S$ ratio across all sites: 5.11; Table 2). Among all the *MSP* genes, only *MSP*5, *MSP*9, and *MSP*10 indicated regions under positive selection. The *MSP*5 and *MSP*9 genes had an average $d_N/d_S$ ratio of 3.85 across all sites and 1.11 across branches (Table 2). While *MSP*10 had an average $d_N/d_S$ ratio of 1.16 across all branches and less than 1 across all sites, only seven sites were indicated with posterior probabilities greater than 50% and 95% (Table 2). Although *MSP*3.8 showed potential positive selection based on the integrated haplotype scores (Fig 4), PAML did not show significant evidence of positive selection.

## Copy number variation and evolution of high-order copy variants

According to CNVnator, six gene regions showed copy number variation among our Ethiopian samples (Fig 5; S5 Table). Among them, three gene regions were detected with up to 2–3 copies and three gene regions with 4 copies or higher. Among the 43 erythrocyte binding genes, duplications were observed in *DBP*1 on chromosome 6 and *MSP*3.11 on chromosome 10. *DBP*1 ranged from one to as high as five copies, and *MSP*3.11 ranged from one to as high as three copies among our samples (Fig 5), consistent with previous findings [19, 20, 79]. The remaining erythrocyte binding genes were detected with a single copy across our samples. Compared to the Ethiopian isolates, a larger number of erythrocyte binding gene regions including *DBP*1, the *MSP*3 and *RBP* gene families were detected with high-order copies in the Southeast Asian isolates (Fig 5; S5 Table). These gene regions showed an average copy number

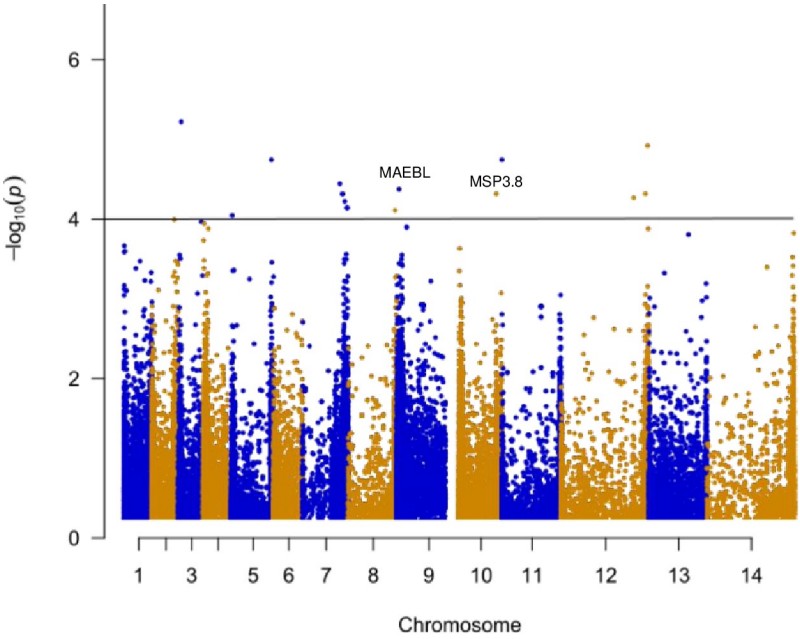

**Fig 4. Signal of positive selection across the 14 chromosomes among all *P. vivax* samples.** Genes that showed significant signal of positive selection included *MAEBL* and *MSP*3.8 gene regions. *MAEBL* is a membrane associated erythrocyte binding like protein that may have a function associated with erythrocyte invasion. Likewise, *PvMSP*3.8 gene may also involve in erythrocyte invasion.

ranged from one to as high as four copies, consistent with previous findings [19]. For instance, 17 of the Southeast Asian samples were detected with 2–3 copies and one sample with 4 or

**Table 2. A shortlist of 17 erythrocyte binding gene candidates that showed signals of positive selection based on the Likelihood Ratio Test of the M1 (neutral model) and M2 models (selection model) in PAML.** Gene ID is presented in S1 Table.

| Gene ID (PlasmoDB) | Gene description | $d_N/d_S$ average across branches (M1) | $d_N/d_S$ average across branches (M2) | $d_N/d_S$ across Sites (M2) | PSS >50% (>95%) M2 Model |
|---|---|---|---|---|---|
| PVP01_0102300 | Duffy binding protein 2/EBP | 0.219 | 2.11 | 6.19 | 9 (8) |
| PVP01_0613400 | rRNA-processing protein ebp2, putative | 0.84 | 1.36 | 2.06 | 5 (5) |
| PVP01_0824100 | microneme associated antigen, putative | 0.09 | 4.97 | 18.4 | 7 (7) |
| PVP01_0418400 | merozoite surface protein 5 | 0.14 | 1.06 | 3.78 | 78 (64) |
| PVP01_1446800 | merozoite surface protein 9 | 0.14 | 1.16 | 3.9 | 33 (31) |
| PVP01_1129100 | merozoite surface protein 10, putative | 0.35 | 1.16 | 0.86 | 7 (7) |
| PVP01_0701200 | reticulocyte binding protein 1a | 0.16 | 1.86 | 4.79 | 28 (24) |
| PVP01_1402400 | reticulocyte binding protein 2a | 0.11 | 1.71 | 5.22 | 18 (15) |
| PVP01_0534400 | reticulocyte binding protein 2 precursor (PvRBP- 2), putative | 0.38 | 1.18 | 3.07 | 96 (44) |
| PVP01_1255000 | rhoptry neck protein 2 | 0.24 | 1.52 | 3.99 | 15 (9) |
| PVP01_0404200 | tryptophan-rich protein | 0.62 | 2.25 | 3.64 | 19 (19) |
| PVP01_0503400 | tryptophan-rich protein | 1 | 11.98 | 21.18 | 6 (1) |
| PVP01_0504200 | tryptophan-rich protein | 0.33 | 1.01 | 2.72 | 3 (3) |
| PVP01_1101400 | tryptophan-rich protein | 0.19 | 1.22 | 4.21 | 5 (5) |
| PVP01_1469800 | tryptophan-rich protein | 0.46 | 2.16 | 6.84 | 10 (10) |
| PVP01_1469900 | tryptophan-rich protein | 0.54 | 2.03 | 6.38 | 2 (2) |
| PVP01_0700800 | tryptophan-rich protein | 0.36 | 1.585 | 4.09 | 1 (1) |

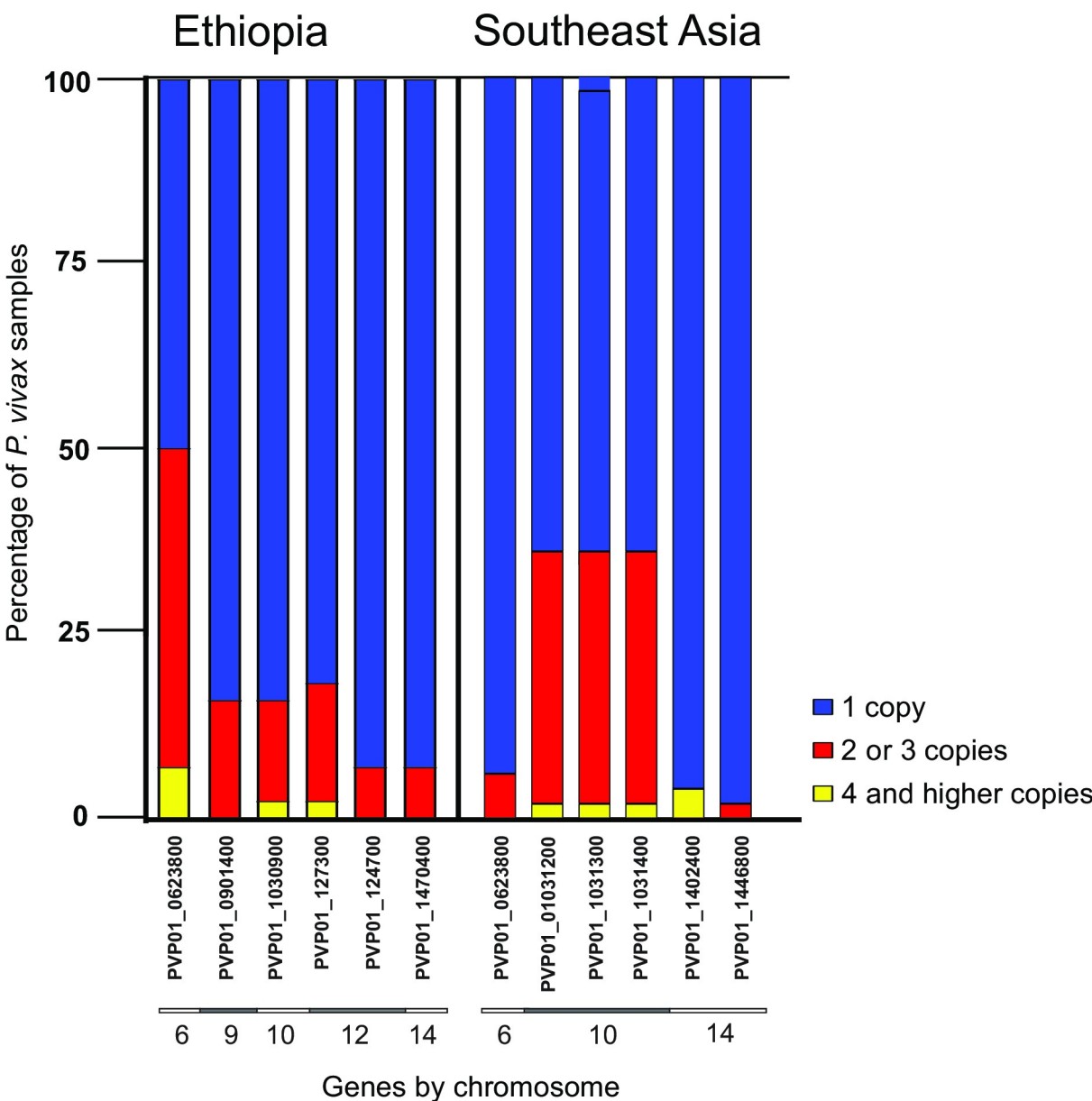

**Fig 5. Gene regions that were detected with copy number variation in the 44 genomes of *P. vivax* from Ethiopia and 50 genomes of *P. vivax* from Southeast Asia.** Annotation of these genes can be found in S4 Table. Among them, *PvDBP*1 (PVP01_0623800) and *PvMSP*3.11 (PVP01_1030900) were associated with erythrocyte invasion. Other genes that were found to have high-order copy number were yet to be described exported plasmodium proteins. Annotation of these genes can be found in S4 Table. Among them, *PvDBP*1 (PVP01_0623800), members of the *PvMSP*3 gene family (PVP01_1031200, PVP01_1031300, PVP01_101400), *PvMSP*9 (PVP01_1445800), and *PvRBP*2a (PVP01_1402400) were associated with erythrocyte invasion. Four or higher copies of *PvMSP*3 and *PvRBP*2a genes were observed in the samples in Southeast Asia samples.

higher copies of *MSP*3.5, *MSP*3.8, and *MSP*3.9. Two samples were detected with four or higher copies of *RBP*2a, and one sample with 2–3 copies of *MSP*9. The higher copy number of *RBP*2a and *MSP*9 were not detected in a previous study by Pearson *et al.* [19] and this could be due to the use of different reference genome, i.e., Sal1 monkey strain in the previous and PVP01 Indonesian patient isolate in the present study. Though fewer erythrocyte binding genes were detected with copy number variation in the isolates from Ethiopia, the range of copy number

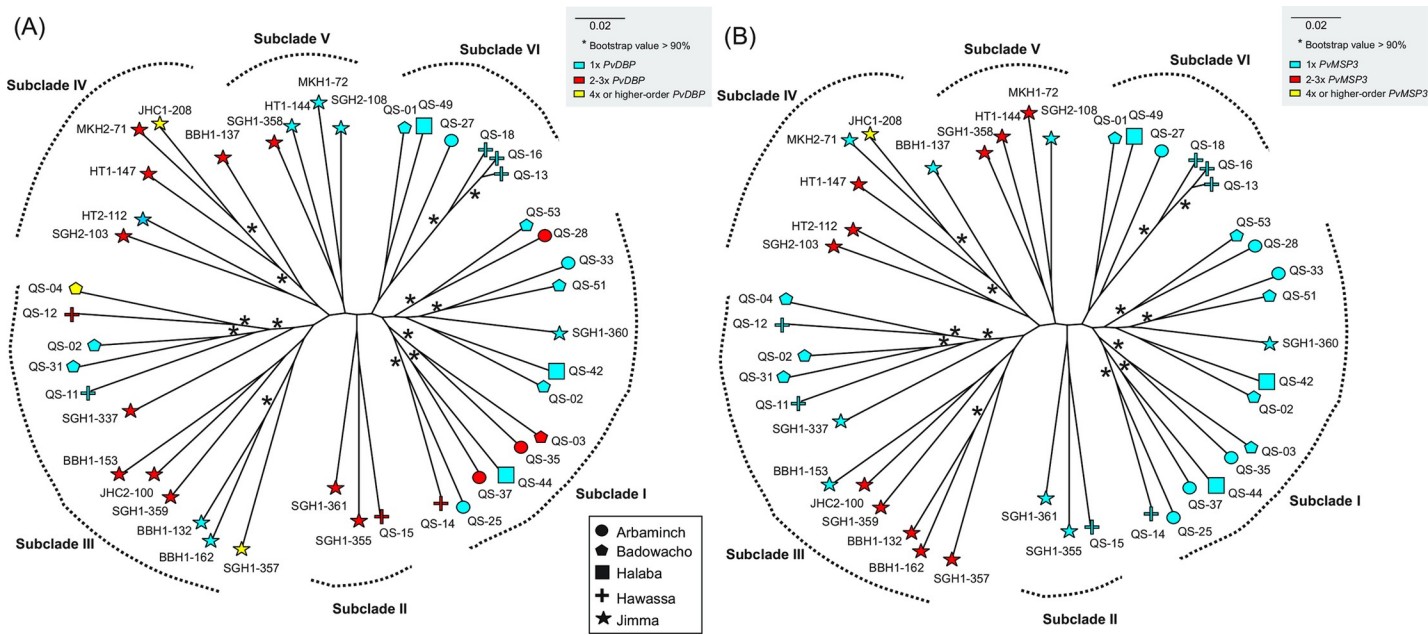

**Fig 6.** An unrooted whole genome phylogenetic tree of the 44 Ethiopian samples showing the evolution of (A) *PvDBP* and (B) *PvMSP*3. The Ethiopian isolates were divided into three subclades. Subclade I contained samples mostly from the Arbaminch (circle) and Badowacho (pentagon). Subclade II contained a mixture of isolates from Arbaminch (circle), Halaba (square), Hawassa (cross), and Jimma (star). Subclade III contained samples from Jimma (star). No distinct clusters were observed between isolates with single and multiple *PvDBP* and *PvMSP*3 genes. These patterns suggest that these gene regions could have expanded multiply among samples at different locations.

e.g., in the *DBP*1 gene was higher in isolates from Ethiopia than in isolates from Southeast Asia (S5 Table).

A maximum likelihood tree constructed based on the whole genome sequences showed an admixture of *P. vivax* isolates with single and multiple *PvDBP* copy number (Fig 6A). The *P. vivax* isolates from Ethiopia were divided into six subclades. Subclade I contained *P. vivax* samples mostly from Arbaminch and Badowacho with both one and two *PvDBP* copies. Subclade II contained samples from Jimma and Hawassa with two *PvDBP* copies. Subclade III contained a mixture of *P. vivax* samples from Arbaminch, Halaba, Hawassa, and Jimma with single and high-order *PvDBP* copies. This clade was sister to subclade IV that contained *P. vivax* samples mostly from Jimma (Fig 6A). In subclade IV, no distinct clusters were detected between isolates with single and multiple *PvDBP*. Subclade V contained samples from Jimma and subclade VI contained samples from Arbaminch, Badowacho, Hawassa, and Halaba. Each of the subclades had samples with both one and two *PvDBP* copies. Similar patterns were observed in the *MSP*3.11 where *P. vivax* isolates with single and multiple copies were clustered together in separate subclades (Fig 6B), suggesting that this gene could have resulted from multiple duplication events.

## Gene flow and transmission network of the Ethiopian *P. vivax*

The $F_{ST}$ statistics showed low levels of genetic differentiation between Arbaminch and Halaba ($F_{ST}$ value = -0.019), as well as between Arbaminch and Hawassa ($F_{ST}$ value = 0.017; S6 Table). By contrast, higher differentiation was observed between Jimma and Arbaminch ($F_{ST}$ value = 0.229) and between Jimma and Halaba ($F_{ST}$ value = 0.327; S6 Table). ADMIXTURE analysis showed that there were most likely two or three genetic clusters among all the Ethiopian samples based on the lowest cross validation scores of 0.75 and 0.78, respectively (Fig 7A).

At $K = 2$, most of the samples from Badowacho, Hawassa, Arbaminch, and Halaba had pre-dominantly the light-blue cluster, whereas samples from Jimma had the red cluster (Fig 7A). At $K = 3$, samples from Badowacho and Hawassa shared a mix of green and blue clusters; samples from Arbaminch had mostly the green cluster; samples from Halaba had mostly the blue cluster, similar to those from Jimma that had predominantly the blue cluster, though red and green clusters were also observed (Fig 7A). This clustering pattern is consistent with the $F_{ST}$ statistics and principal component analysis (S1 Fig), which showed the greatest genetic differences and lessen gene flow between Jimma and other study sites.

The transmission network indicated that Arbaminch was the major source of infections from which the infections in Jimma, Hawassa, Badowacho, and Halaba originated (Table 3; Fig 7B). In contrast, no transmission originated from Halaba, making this location the largest sink of transmissions. The greatest extent of transmission was observed between Arbaminch and Badowacho (Fig 7B). Hawassa and Jimma showed a source hub ratio of 0.5, indicating that there are equally as many egress transmissions as ingress transmissions (Table 3). Although Jimma and Badowacho/Halaba are in close geographical proximity, no apparent gene flow was observed between these sites.

## Discussion

Across the genome, the total number of SNPs observed among 44 *P. vivax* isolates from Ethiopia were lower than those previously reported for *P. vivax* isolates from South American [80] and Southeast Asian countries, despite the different analytical tools used for SNP calling [19]. For instance, 303,616 high-quality SNPs were detected in 228 *P. vivax* isolates from Southeast Asia and Oceania in a previous study, of which Sal-I was used as the reference sequence and subtelomeric regions were discarded [19]. Auburn *et al.* [20] found that the average nucleotide diversity of *P. vivax* isolates from Ethiopia was lower than *P. vivax* isolates from Thailand and Indonesia, but higher than *P. vivax* isolates from Malaysia.

Chromosomes 3, 4, and 5 have been previously shown to contain the lowest proportion of synonymous SNPs than the other parts of the genome [12]. In the present study, chromosomes 3 and 6 were found to have the lowest number of both synonymous and nonsynonymous SNPs. This follows observations made in other studies done with nucleotide diversity ranging from 0.8 SNPs per kb in North Korea to 0.59 SNPs per kb in Peru [81]. Among the 64 erythrocyte binding gene candidates, the MSP and RBP multigene families showed the highest level of genetic variation. This agrees with previous studies that reported a remarkably high diversity in *RBP*2 compared to *RBP*1 and its homolog group in *P. falciparum* [31]. In the Greater Mekong Subregion, the *MSP*3 and *PIR* gene families also indicated high levels of genetic diversity with 1.96% and 1.92% SNPs per base respectively, confirming that members of multigene families are highly variable genetically [30, 82]. Such diversity suggested that the binding domains of these genes could be under differential selection pressure. This pattern has been observed in previous studies and is likely due to their critical role in reticulocyte invasion, immunogenic properties, and human migration [26, 83–85].

For the *P. vivax* isolates from Southeast Asia, copy number variation was observed previously in *DBP*1, ranging from 1 to 2–3 copies using the Sal1 reference genome [19]. In the present study, copy number variation was detected in six erythrocyte binding gene regions including *DBP*1, and members of the *MSP*3 and *RBP* gene families using the PVP01 reference genome. However, we found no evidence of duplications in the *RBP* gene family in the isolates form Ethiopia. Among the isolates from Ethiopia, the highest and most variable copy number variations were detected in the *DBP*, with copy numbers ranging from one to as high as five. Likewise, for the *MSP*3, copy numbers ranging from one to as high as four. Based on the

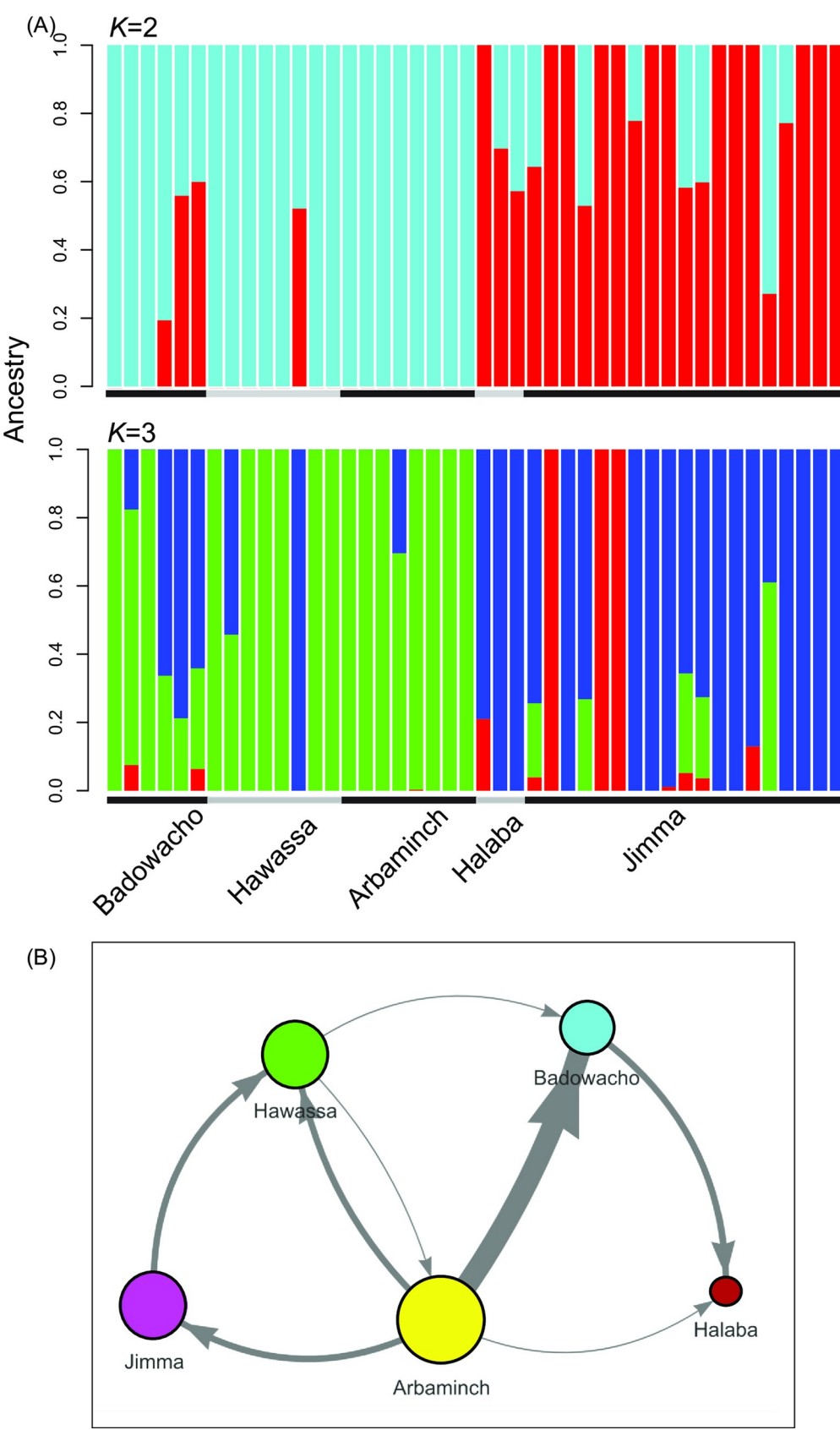

**Fig 7.** (A) Bar plot based on ADMIXTURE analyses of SNP variants showing the clustering pattern of samples among study sites in Ethiopia. Two or three genetic clusters were identified based on the cross-validation scores. At *K* = 2, most of the samples from Badowacho, Hawassa, Arbaminch, and Halaba had predominantly the light-blue cluster, whereas samples from Jimma had the red cluster. At *K* = 3, samples from Badowacho and Hawassa shared a mix of green and blue clusters; samples from Arbaminch had mostly the green cluster; samples from Halaba had mostly the blue cluster, similar to those from Jimma that had predominantly the blue cluster, though red and green clusters were also observed. (B) The transmission network, created using the StrainHub program, indicated that Arbaminch was the major source of infection in Jimma, Halaba, Badowacho and Hawassa. The greatest extent of transmission (indicated by the boldest arrow) was observed between Arbaminch and Badowacho. Even though Jimma, Badowacho and Halaba are geographically in close proximity, transmission was not intense among these sites.

phylogeny, *DBP* and *MSP*3 expansion had occurred multiple times as tandem copies. These findings were consistent with earlier studies [19, 79] and suggested that gene expansion may play a key role in host cell invasion [86]. Despite a higher malaria transmission intensity in Southeast Asia than in Ethiopia, *PvDBP* only ranged from 1 to 2–3 copies among the Southeast Asian isolates, much less than that observed among the isolates from Ethiopia. The range of *PvDBP* copy variation may not be related to transmission intensity but host-parasite interaction during the invasion process. Given that Duffy phenotype is more diverse among the the human populations in Ethiopia than in Asia [87], it is possible that *P. vivax* expands *DBP* to multiple copies to enhance binding affinity and invasion to different host phenotype [79]. These expanded gene copies shared identical nucleotide sequences, suggesting rapid and recent duplications within an isolate [79, 88]. Our phylogenetic analyses indicated that *P. vivax* with high-order *DBP* copies evolved from isolates with a single-copy *DBP* multiple times independently at different geographic locations. Isolates with high-order *PvDBP* copies may confer better fitness and favored by selection, resulting in a higher frequency in isolates *of P. vivax* from Ethiopia. Moreover, gene flow among geographical locations may allow further spread and a broad distribution of high-order *PvDBP* strains. For all other putative erythrocyte binding genes, only a single copy was detected among all samples. A larger sample in future investigations would verify this observation.

In the present study, we identified a panel of 43 putative erythrocyte binding gene candidates based on the information from the literature and multiple databases. We are currently validating the functions for each of these genes by binding assays based on the gene variants observed in the isolates from Ethiopia. Among these 43 putative erythrocyte binding gene candidates, *MAEBL* was shown to be highly conserved in *Plasmodium* [89] and had a significant signal for positive selection among the *P. vivax* samples from Ethiopia, agrees with the findings of a recent study [89]. In *Plasmodium berghei*, *MAEBL* is a sporozoite attachment protein that plays a role in binding and infecting the mosquito salivary gland [90]. In *Plasmodium falciparum*, *MAEBL* is found in the rhoptries and on the surface of mature merozoites and expresses at the beginning of schizogony [90]. In *P. vivax*, *MAEBL* is a conserved antigen expressed in blood stages, as well as in the mosquito midgut and salivary gland sporozoites [90, 91]. The *MAEBL* antigen contains at least 25 predicted B-cell epitopes that are likely to elicit antibody-

**Table 3. Transmission network metrics among study sites calculated by StrainHub.**

| Metastates | Degree Centrality | Indegree Centrality | Outdegree Centrality | Betweenness Centrality | Closeness Centrality | Source Hub Ratio |
|---|---|---|---|---|---|---|
| Arbaminch | 5 | 1 | 4 | 2 | 0.25 | 0.8 |
| Badowacho | 3 | 2 | 1 | 1 | 0.2 | 0.33 |
| Halaba | 2 | 2 | 0 | 0 | 0.17 | 0 |
| Hawassa | 4 | 2 | 2 | 3 | 0.2 | 0.5 |
| Jimma | 2 | 1 | 1 | 0 | 0.17 | 0.5 |

dependent immune responses [92]. Thus, positive selection observed in this gene region among the *P. vivax* isolates from Ethiopia could be associated with the immunity-mediated selection pressure against blood-stage antigens. Apart from *MAEBL*, positive selection was also detected in the *MSP3* gene among the *P. vivax* isolates from Ethiopia and may have important implications on the susceptibility of human hosts [93]. In *P. vivax*, the *MSP3* gene and its paralogs such as *MSP3-alpha* and *MSP3-beta* on chromosome 10 have been shown to simultaneously express in the blood stage merozoite and are immunogenic [26]. These paralogs may have functionally redundant roles in determining antigenicity [26]. Extensive polymorphisms have been reported throughout the gene family, likely through frequent recombination and gene conversion between the *PvMSP*3 paralogs [26, 94]. The central low complexity regions of *PvMSP*3, where indels and high level of polymorphisms were observed, are highly immunogenic compared to the more conserved N-C terminals [93, 95]. Thus, selection for SNP variants in *PvMSP*3 and significant expansion of the gene family in the *P. vivax* isolates from Ethiopia may relate to the increased capability of the red blood cell invasion process and immune evasion [96, 97], though no significant selection was detected in other geographical isolates [19, 89]. While *DBP*1 had the highest and most diverse copy number variation, no significant signal of positive selection was detected, consistent with the findings in *P. vivax* isolates from from Western Cambodia, Western Thailand, and Papua Province in Indonesia [19, 20]. A number of antimalarial drug resistance genes including the chloroquine resistance transporter (*CRT-O*) gene, the dihydropteroate synthase (*DHPS*) gene, and dihydrofolate reductase–thymidylate synthase (*DHFR*) gene were previously reported with positive selection [20, 89], but we did not detect such in this study. Sequences of *PvCRT-O* and *PvMDR*1 have been shown to be highly conserved in *P. vivax* isolates from Ethiopia [53]. Broader samples are needed to further examine the role of these genes in chloroquine resistance.

It is noteworthy that the calculation of integrated haplotype scores and the accuracy of phasing genotypes using BEAGLE were dependent on the levels of linkage disequilibrium of the whole genomes. The higher the levels of linkage disequilibrium, the more accurate are the phased genotypes and thus the iHS score. Pearson *et al.* [19] found that *P. vivax* experienced drops in linkage disequilibrium after correcting for population structure and other confounders. Linkage disequilibrium of *P. vivax* genomes has been previously shown to be associated with the rate of genetic recombination and transmission intensity [98–100]. In high transmission sites of Papua New Guinea and the Solomon Islands, no identical haplotypes and no significant multilocus LD were observed, indicating limited inbreeding and random associations between alleles in the parasite populations [101, 102]. However, when transmission intensity declined, similar haplotypes and significant LD were observed possibly due to self-fertilization, inbreeding and/or recombination of similar parasite strains [98]. Multilocus LD is significantly associated with the genetic relatedness of the parasite strains [103], but inversely associated with the proportion of polyclonal infections [104]. In Southwestern Ethiopia, malaria transmission ranged from low to moderate, and LD levels varied markedly among the study sites [53, 105]. To address this limitation in BEAGLE, all genes that were detected with positive selection in BEAGLE were further analyzed with PAML for verification. Future study should include broad samples to thoroughly investigate selection pressure at the population level and the function significance of polymorphisms in the *MSP3* genes.

Previous studies have shown high levels of genetic diversity among *P. vivax* isolates from endemic countries, such as Papua New Guinea, Cambodia, and Myanmar [16, 106, 107]. Such a diversity was directly related to high transmission intensity and/or frequent gene exchange between parasite populations via human movement [4, 12, 13, 53]. For example, previous studies using microsatellites have demonstrated a consistently high level of intra-population

diversity ($H_E$ = 0.83) but low between-population differentiation ($F_{ST}$ ranged from 0.001–0.1] in broader regions of Ethiopia [53, 105]. High heterozygosity was also observed in *P. vivax* populations from Qatar, India, and Sudan (average $H_E$ = 0.78; 62), with only slight differentiation from *P. vivax* in Ethiopia ($F_{ST}$ = 0.19) [108]. Frequent inbreeding among dominant clones [98, 101] and strong selective pressures especially in relapse infections [19, 20, 108, 109] may also contribute to close genetic relatedness between and within populations. Thus, in this study, it is not surprising to detect a high level of parasite gene flow among the study sites at a small geographical scale, despite the limited number of samples. In the present study, we successfully employed a transmission network model to identify transmission paths, as well as the source and sink of infections in the region, beyond simply indicating genetic relationships. The genetic differences observed between Jimma and the other sites could be explained by the amount of human movements. Sites such as Arbaminch and Hawassa in Ethiopia are well known sight-seeing areas famous of their natural scenery and heritage sites that attract large number of local and foreign tourists every year, whereas Jimma is a local residential area where fewer people from outside visit.

To conclude, this study elaborates the genomic features of *P. vivax* in Ethiopia, particularly focusing polymorphisms in erythrocyte binding genes that potentially play a key role in local parasite invasion, a critical question given the mixed Duffy positive and negative populations of Ethiopia. The findings provided baseline information on the genomic variability of *P. vivax* infections in Ethiopia and allowed us to compare the genomic variants of *P. vivax* between Duffy-positive and Duffy-negative individuals as the next step of our ongoing investigation. Further, we are in progress of developing a panel of informative SNP markers and using them on larger sample sets to track transmission at a micro-geographical scale.

## Supporting information

**S1 Table. Distribution of SNP variants in the 43 *P. vivax* erythrocyte binding gene candidates among the 44 Ethiopian genomes.**
(DOCX)

**S2 Table. Distribution of single nucleotide polymorphism (SNP) variants across the 14 *P. vivax* chromosomes of the 44 Ethiopian genomes.**
(DOCX)

**S3 Table. $F_{ws}$ statistics for all 44 Ethiopian samples calculated using the moimix package in R.**
(DOCX)

**S4 Table. Likelihood Ratio Test results of the M1 (neutral model) and M2 models (selection model) in PAML of all the 43 erythrocyte binding gene candidates.**
(DOCX)

**S5 Table.** (A) Gene regions that were detected with copy number variation among the 44 Ethiopian *P. vivax* isolates based on CNVnator. Among them, only two erythrocyte binding gene candidates *PvDBP*1 and *PvMSP*3 were detected with high-order copies. (B) Gene regions that showed copy number variation among the 50 Southeast Asian isolates based on CNVnator. Among them, six gene regions were detected with gene duplication and four had high-order copies.
(DOCX)

**S6 Table. $F_{ST}$ values calculated between study sites using the VCFTools procedure.**
(DOCX)

**S1 Fig. Principal component analysis plot based on the SNP information from our variant analysis.** Samples obtained from Jimma were clustered together, whereas samples from Arbaminch, Badowacho, Hawassa, and Halaba were mixed, with the exception of two samples from Hawassa. This clustering pattern suggested that there was considerable genetic variation among study sites even at a small geographical scale.
(TIF)

## Acknowledgments

We are greatly indebted to the staff and technicians from Jimma University for field sample collection, the communities and hospitals for their support and willingness to participate in this research. We thank various units at the University of North Carolina at Charlotte including: the Department of Biological Sciences, the Department of Bioinformatics and Genomics, and the Department of Mathematics.

## Author Contributions

**Conceptualization:** Anthony Ford, Julian C. Rayner, Guiyun Yan, Delenasaw Yewhalaw, Eugenia Lo.

**Formal analysis:** Anthony Ford, Richard Pearson.

**Funding acquisition:** Julian C. Rayner, Guiyun Yan, Eugenia Lo.

**Investigation:** Anthony Ford, Daniel Kepple, Eugenia Lo.

**Methodology:** Anthony Ford, Beka Raya Abagero, Julian C. Rayner, Eugenia Lo.

**Project administration:** Delenasaw Yewhalaw, Eugenia Lo.

**Resources:** Beka Raya Abagero, Sisay Getachew, Delenasaw Yewhalaw, Eugenia Lo.

**Software:** Anthony Ford, Daniel A. Janies.

**Supervision:** Daniel A. Janies, Eugenia Lo.

**Validation:** Anthony Ford, Sarah Auburn, Karthigayan Gunalan, Louis H. Miller, Eugenia Lo.

**Visualization:** Anthony Ford, Daniel Kepple, Colby Ford, Eugenia Lo.

**Writing – original draft:** Anthony Ford, Daniel Kepple, Jordan Connors, Eugenia Lo.

**Writing – review & editing:** Anthony Ford, Daniel Kepple, Karthigayan Gunalan, Louis H. Miller, Daniel A. Janies, Julian C. Rayner, Eugenia Lo.

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
