## [Decision Letter · Decision Letter 0]

21 Apr 2020

Dear Mr. Ford,

Thank you very much for submitting your manuscript "Whole Genome Sequencing of Plasmodium vivax Isolates Reveals Frequent Sequence and Structural Polymorphisms in Erythrocyte Binding Genes" for consideration at PLOS Neglected Tropical Diseases. As with all papers reviewed by the journal, your manuscript was reviewed by members of the editorial board and by several independent reviewers. In light of the reviews (below this email), we would like to invite the resubmission of a significantly-revised version that takes into account the reviewers' comments. 

We cannot make any decision about publication until we have seen the revised manuscript and your response to the reviewers' comments. Your revised manuscript is also likely to be sent to reviewers for further evaluation.

Sincerely,

Rhoel Ramos Dinglasan

Associate Editor

Ana Rodriguez

Deputy Editor

Reviewer's Responses to Questions

**Key Review Criteria Required for Acceptance?**

**Methods**

-Are the objectives of the study clearly articulated with a clear testable hypothesis stated?

-Is the study design appropriate to address the stated objectives?

-Is the population clearly described and appropriate for the hypothesis being tested?

-Is the sample size sufficient to ensure adequate power to address the hypothesis being tested?

-Were correct statistical analysis used to support conclusions?

-Are there concerns about ethical or regulatory requirements being met?

Reviewer #1: The methods and analysis as they are, are sufficient to understand what experiments and computational work was done. But many details are missing that make the conclusions difficult to understand.

1. The read depth from all genomes seems in conflict with the analyses done. Line 234 says a read depth of >30 was required for all SNPs, but Table 1 indicates about 22 samples with an average read depth less than 30.

2. I'm not quite sure how clonality was determined. I don't see the results of the microsatellite experiments. Additionally, were the publicly available data from monoclonal infections? If they weren't then many of these analyses might be difficult to understand.

Reviewer #2: (No Response)

**Results**

-Does the analysis presented match the analysis plan?

-Are the results clearly and completely presented?

-Are the figures (Tables, Images) of sufficient quality for clarity?

Reviewer #1: I think these studies are missing a critical outgroup of vivax genomes from a different area. (eg, among the genes that are under positive selection, are they just under positive selection in this dataset from Ethiopia, or is it observed elsewhere?) Additionally, are there differences in SNPs between Duffy-negative and Duffy-positive patients?

What is the explanation for the DBP CNV is different geographic locations? If I am understanding correctly, Fig 6 shows relatively low transmission between sites, yet multiple DBP copies are found in a few different sites? Can the associated SNPs explain how this came about?

Line 334 - These numbers seem significant, but shouldn't the size of the chromosome be taken into account? Maybe some statistics (expected # if SNPs were randomly distributed)

Line 347 - What does the lowest number mean? Among all annotated genes, or just the erythrocyte binding genes?

Table 1 - Samples generated in this study should be indicated in some way.

Figure 4 - I think this should include gene names.

Figure 5 - I think where these samples came from is an important part of the interpretation of this figure and is missing.

Figure 6 - I don't think a 3D PCA is required here.

Reviewer #2: (No Response)

**Conclusions**

-Are the conclusions supported by the data presented?

-Are the limitations of analysis clearly described?

-Do the authors discuss how these data can be helpful to advance our understanding of the topic under study?

-Is public health relevance addressed?

Reviewer #1: I think the conclusions are supported by the data, but it is difficult to determine if the gene flow/evolution studies are unique to Ethiopia from this work as is. And I think a greater discussion of how this data can be used to understand Duffy-negative infections is missing.

Line 484 - How did you validate these genes? I think this is an important point because as is, the entire focus of the manuscript is on evolution in genes thought to be related to invasion, so this validation is important.

Line 497 - Are there SNPs between the copy numbers? ie are all DBPs in the same genome identical?

Reviewer #2: (No Response)

**Editorial and Data Presentation Modifications?**

Reviewer #1: 116 - This sentence needs editing

123 - These genes should all be in a consistent PvP01 gene name

Reviewer #2: (No Response)

**Summary and General Comments**

Reviewer #1: The results presented here do not match the hypothesis or question initially introduced. The introduction sets up the paper as being about SNPs or CNVs related to vivax malaria among Duffy-negative individuals. And choosing this region of Ethiopia, with increasing rates of vivax malaria and Duffy-negative individuals, makes this kind of genomic study very exciting and an important dataset. But this question is not addressed in any of the analyses. 

Going forward, two possible analyses could be performed to better place the data within the initial question. 1) Add an analysis where the Duffy-positive and Duffy-negative individuals are compared. It's possible though that this analysis would be too difficult and beyond the scope of the paper because A) 24 of the publicly available samples are unknown in regards to Duffy, B) because transmission in this region, Duffy-negative and Duffy-positive patients may still have active transmission between each other, which would be confounding. So an alternative analysis would be: 2) Add an analysis with other vivax genomes from other regions. I understand this is similar to "Auburn et al 2019," but it is difficult to interpret these results without knowing how these vivax compare to vivax in other regions where Duffy-negative infections are not happening.

Reviewer #2: The manuscript described 20 new P. vivax genome sequences from Ethiopia and performed population genomics analysis of these samples with 24 samples from the same region published earlier. While the overall study appears to be sound, more emphasis may need to be on the populations in Ethiopia, and whether more detailed SNP analysis from WGS provides better resolution of the parasite populations in this country than the studies done using microsatellite markers. If this is true, a panel of SNPs that can be used to better reflect parasite population divisions in this country would indeed be very useful. However, just from reading the abstract, these important issues are not reflected in the abstract. 

Major comments

1. Authors claimed that these Pv sequences “allallowed us to develop a panel of informative Single Nucleotide Polymorphic markers diagnostic at a micro-geographical scale”. This would be a nice to identify a panel of SNPs that can be used to distinguish parasites at micro- scale. However, I did not see anywhere about this panel of SNPs. This would be interesting given the divergence of Jimma parasites from other parts.

2. The authors detect clonality using 7 microsattelite markers, then claim regarding the WGS “Similarly, we concluded that the percentage of the P. vivax reads covered in the additional 24 samples were high enough to reflect the dominant signal of the variants and negate polyclonal influences.” From what I could tell there’s no discussion of how they came to this conclusion. Analyses such as non-reference allele distributions and Est-MOI are very easy to perform with WGS data and would confirm clonality. 

3. Bowtie was used to analyze the 22 samples and call for SNPs, whereas BWA-MEMv.2 was used to align the 24. I wonder why not doing all alignments using the same program, since two programs with different parameter settings may give you different results. Could this be the cause of the significant divergence of Jimma samples from the rest since SNP calling used different methods?

4. Strainhub appears to use a relatively new methodology which uses phylogenetic tree state changes to map closeness. It would be good to use additional methods to confirm some of the findings regarding population sharing and geneflow; i.e. IBD sharing could help with high/low transmission regions, Fst could be used to complement genetic isolation findings. Additionally, I’d like to hear the authors comment on whether or not their analysis is sensitive to sample size, as the authors of the programs commented in their paper that “the results are only as strong as the underlying phylogenetic data (sampling across metadata states and taxa…)” and this could factor into their identification of Halaba, which only has 3 samples, as a sink. This makes us feel that the source-sink analysis is not robust. 

5. I suggest that the authors use additional tests such as Structure to compare the relationships among the parasites from different regions. Together with phylogeny and PCA, this may provide further corroboration of the data.

6. I think the authors try to analyze the RBC invasion related genes in order to identify genes potentially involved in the invasion of Duffy negative RBC. However, this part is flawed from the use of genes that are apparently involved in RNA metabolism. The authors analyzed a DBP family of genes because these genes carry an abbreviation of DBP. Two of the genes included in this family are the Duffy binding protein (DBP), whereas the rest are simply proteins of the DEAD box RNA helicases – DEAD box protein (which happen to be abbreviated as DBP). But they have nothing to do with invasion.

7. The pir gene copy variation part may not be correct and undermines their conclusions. PIR genes are really common sources of mismapping. I don’t see any specific actions to control for this in the data. On line 458 authors suggest that using P01 mitigates this because its subtelomeric ends are more complete, but to my understanding a lot of the reason mismapping occurs on these genes is because the repeat regions are highly repetitive and the genes are highly variable, meaning it’s very easy for algorithms to misidentify the best match. Because of the high coverage variability this could affect copy number predictions from CNVnator. Additionally, because of misaligned reads it could affect SNP-calling—a problem that can also come from duplications. It may be worth doing a supplemental analysis specifically on high interest PIR genes to confirm mismapping wasn’t a problem either using read visualization or PCR. This may similarly to other claims of the msp3 gene family amplifications.

Minor comments:

Sample collection part is confusing. Line 184-196: This part described collection of 22 Pv clinical samples in 2016 and DNA extracted for sequencing. Then they further described in lines 197-201 that another larger set of samples were genotyped and used for sequencing. I am confused how many from each place or sample set? The line 203, “The original 22 samples were processed to remove reads other than P. vivax”, which original, the 22 samples from Jimma in 2016? How about the larger set of samples? Then they extracted WGS from additional 24 samples – please add a reference showing that these are previously published data. 

Line 208: “the Duffy status of each of these 24 samples is not known”. I assume from this statement that the rest of 22 samples have known Duffy status. If this is the case, would that be useful to compare the parasite isolates from Duffy negative with those from Duffy positive hosts (even though the number is small)?

Line 186: malaria symptomatic (who……..), to me this is too early since you collected blood for microscopy (next sentence). For this sentence, I would use those with symptoms suspected of malaria infection.

Line 227: “3,000 million” > 3000? 3million? 3billion?

Line 343: I wonder if the authors provide somewhere backgrounds for msp3 gene family (how these new nomenclatures correspond to many references describing msp3alpha, beta and gamma)?

Line 379 (and elsewhere): the members included are not all DBP family proteins.

Line 386-390: Pir copy number part needs to be removed (see reasons above)

Line 392: duplications were observed for msp3 on chromosome 10. Do you consider all the msp3 gene family members as duplications? Or a particular member. Actually many msp3 gene members are suspected to have arisen from past duplication events and then divergence, since their sequences differ dramatically.

Figure 6A is useful and may be better to include collection sites information. Figure 6B-D do not seem to provide more information.

Line 411: “could have expanded multiply among samples at different locations.” > could have resulted from multiple duplication events

Line 429: I’m not sure it’s a good idea to compare to reference 19 for blanket number of SNPs because you used two different pipelines to call them. 

Line 441: “remarkably high diversity in RBP2 than in RBP1” > remarkably high diversity in RBP2 compared to RBP1

Line 423-424: Given the size of Ethiopia and the close proximity of the collection sites, it is hard to understand why the lack of gene flow? This deserves some discussion (any gene flow barriers, different ethnicities? …….)

PLOS authors have the option to publish the peer review history of their article (what does this mean?). If published, this will include your full peer review and any attached files.

Reviewer #1: No

Reviewer #2: No
---

## [Decision Letter · Decision Letter 1]

21 Aug 2020

Dear Mr. Ford,

We are pleased to inform you that your manuscript 'Whole Genome Sequencing of Plasmodium vivax Isolates Reveals Frequent Sequence and Structural Polymorphisms in Erythrocyte Binding Genes' has been provisionally accepted for publication in PLOS Neglected Tropical Diseases.

Best regards,

Rhoel Ramos Dinglasan

Associate Editor

Ana Rodriguez

Deputy Editor

Reviewer's Responses to Questions

**Key Review Criteria Required for Acceptance?**

**Methods**

-Are the objectives of the study clearly articulated with a clear testable hypothesis stated?

-Is the study design appropriate to address the stated objectives?

-Is the population clearly described and appropriate for the hypothesis being tested?

-Is the sample size sufficient to ensure adequate power to address the hypothesis being tested?

-Were correct statistical analysis used to support conclusions?

-Are there concerns about ethical or regulatory requirements being met?

Reviewer #1: (No Response)

**Results**

-Does the analysis presented match the analysis plan?

-Are the results clearly and completely presented?

-Are the figures (Tables, Images) of sufficient quality for clarity?

Reviewer #1: The addition of data from SE Asia in Figures 3 and 4 are incredibly helpful and interesting. These figures increase the confidence in the rest of the paper significanty.

**Conclusions**

-Are the conclusions supported by the data presented?

-Are the limitations of analysis clearly described?

-Do the authors discuss how these data can be helpful to advance our understanding of the topic under study?

-Is public health relevance addressed?

Reviewer #1: (No Response)

**Editorial and Data Presentation Modifications?**

Reviewer #1: (No Response)

**Summary and General Comments**

Reviewer #1: The changes made were very good. Most of my questions were answered and edited in the manuscript to explain the data and impact of it.

I think the track changes version of the manuscript was not uploaded correctly as I can see only one section of formatting changes, and had to find all of the new edits myself.

PLOS authors have the option to publish the peer review history of their article (what does this mean?). If published, this will include your full peer review and any attached files.

Reviewer #1: No

---

## [Editor Report · Acceptance letter]

1 Oct 2020

Dear Mr. Ford,

We are delighted to inform you that your manuscript, "Whole Genome Sequencing of *Plasmodium vivax* Isolates Reveals Frequent Sequence and Structural Polymorphisms in Erythrocyte Binding Genes," has been formally accepted for publication in PLOS Neglected Tropical Diseases.

Best regards,

Shaden Kamhawi

co-Editor-in-Chief

Paul Brindley

co-Editor-in-Chief
